# Molecular Modeling of Allosteric Site of Isoform-Specific Inhibition of the Peroxisome Proliferator-Activated Receptor PPARγ

**DOI:** 10.3390/biom12111614

**Published:** 2022-11-01

**Authors:** Suliman Almahmoud, Haizhen A. Zhong

**Affiliations:** 1Department of Medicinal Chemistry and Pharmacognosy, College of Pharmacy, Qassim University, Buraidah 51452, Saudi Arabia; 2Department of Chemistry, University of Nebraska at Omaha, Omaha, NE 68182, USA

**Keywords:** PPAR gamma, anticancer, prostate cancer, docking, allosteric, orthosteric, and selectivity

## Abstract

The peroxisome proliferator-activated receptor gamma (PPARγ) is a nuclear receptor and controls a number of gene expressions. The ligand binding domain (LBD) of PPARγ is large and involves two binding sites: orthosteric and allosteric binding sites. Increased evidence has shown that PPARγ is an oncogene and thus the PPARγ antagonists have potential as anticancer agents. In this paper, we use Glide Dock approach to determine which binding site, orthosteric or allosteric, would be a preferred pocket for PPARγ antagonist binding, though antidiabetic drugs such as thiazolidinediones (TZDs) bind to the orthosteric site. The Glide Dock results show that the binding of PPARγ antagonists at the allosteric site yielded results that were much closer to the experimental data than at the orthosteric site. The PPARγ antagonists seem to selectively bind to residues Lys265, Ser342 and Arg288 at the allosteric binding site, whereas PPARγ agonists would selectively bind to residues Leu228, Phe363, and His449, though Phe282 and Lys367 may also play a role for agonist binding at the orthosteric binding pocket. This finding will provide new perspectives in the design and optimization of selective and potent PPARγ antagonists or agonists.

## 1. Introduction

Peroxisome proliferator-activated receptors (PPARs) are nuclear receptors that function as transcription factors upon activation, regulating as lipid and glucose metabolism and cellular differentiation [1,2,3]. PPAR family is consisted of three isoforms, PPARα, PPARγ, and PPARδ; each isoform has different tissue distribution, selectivity, and responsiveness to ligands [1,4]. PPARγ is an interesting target because of its association with disorders such as atherosclerosis, diabetes, obesity, and cancer [5]. The thiazolidinedione (TZD) drugs such as rosiglitazone and pioglitazone are classical PPARγ agonists and have been used as antidiabetic drugs to treat type 2 diabetes [6]. However, increased PPARγ expression has been found to control other pathways that could induce cancer development and progression. Extended pioglitazone use increased the risk of bladder cancer in a dose- and time-dependent manner [7]. However, a meta-analysis of randomized clinical trials with rosiglitazone and control did not confirm a significant risk of cancer in patients with rosiglitazone treatment. The effects of rosiglitazone on incident malignancies were 0.88 in gastro-intestinal (GI), 0.67 in pulmonary, 1.19 in mammary, and 1.02 in prostate, which suggests some protective effect of rosiglitazone use in GI and lung cancer patients but negative effect in breast and prostate cancer patients [8]. Nevertheless, rosiglitazone was associated with a significantly increased risk of myocardial infarction and death from cardiovascular causes [9]. 

PPARγ was once considered to be a tumor suppressor, and thus PPARγ agonists such as troglitazone were used to treat prostate cancer in clinical trials [10,11,12,13]. However, further analysis showed that the observed inhibition of prostate cancer growth by these compounds was mediated by PPARγ-independent mechanisms such as the inhibition of the CXCR4/CXCL12 axis, or the Bcl-xL/Bcl-2 functions or the suppression of the androgen receptor expression [14,15,16]. In fact, an increased level of PPARγ expression was observed in prostate cancer and was positively correlated with stage and grade of prostate cancer, and thus PPARγ has been considered as an oncogene [17,18,19,20]. Overexpression of PPARγ in androgen receptor-negative prostate cancer cell lines such as DU-145, PC3, and PC3M, promoted cell proliferation, migration levels and metastases to the lungs and lymph nodes [17]. siRNA knockdown of PPARγ and treatment of PPARγ antagonist (warfarin) led to reduced prostate cancer tumor size and thus inhibited prostate cancer cell growth [17,21]. Via virtual screening and biological screening, we have identified seventeen hit molecules with single digit micromolar concentration in terms of growth inhibition of PPARγ activity in the LNCaP cells [22]. Therefore, PPARγ antagonists could be used to treat prostate cancer.

In addition, PPARγ antagonists were shown to inhibit the growth of bladder, breast, pancreatic, and prostate cancer cells [23,24,25,26]. Several PPARγ antagonists (such as SR1664) have also yielded therapeutic effects in the treatment of obesity with better therapeutic index comparing with PPARγ agonists by overcoming thiazolidinedione drugs’ undesirable side effects, such as heart failure, and fluid retention with increased risk of weight gain, loss of bone mineral density [27,28]. Therefore, it is highly desirable to design PPARγ antagonists. They have potentials to be anticancer or antidiabetic drugs. 

The PPARγ structure constitutes the N-terminal domain co-regulator-binding activation function 1 (AF1), a central DNA-binding zinc-finger domain, and a C-terminal domain, which includes ligand-binding domain and activation function 2 (AF-2) for a second co-regulator. There is a hinge region that connects the ligand-binding domain and DNA-binding domain [29,30]. The C-terminal domain plays a vital role in ligand binding, dimerization, and transactivation functions [29]. The crystal structure (PDB ID, 2HFP, a human PPARγ/ligand complex) of ligand binding domain of PPARγ consists of 13 α-helices and four β-sheet strands. The PPARγ shares similar ligand binding domain with other nuclear-receptor structures from helix 3 to carboxy-terminal domain, but it has a longer helix H2b [29]. The PPARγ ligand binding domain has a large, hydrophobic, and T-shaped cavity which allows it to accommodate two ligands. One is for orthosteric binding, the other for allosteric binding (Figure 1). The orthosteric binding domains that thiazolidinedione drugs such as rosiglitazone and pioglitazone bind lie in the pocket defined by helices 5, 7, and 11, whereas the alternate binding site is consisted of helices H2b, H3 and the β-sheets (an allosteric site) [31]. 

TZDs are PPARγ agonists that initiate transcription through binding to a canonical orthosteric pocket (Figure 1). Residues for agonist-binding include Cys285, Ser289, His323, Tyr327, Lys367, His449, and Tyr473. In addition to orthosteric binding, several PPARγ ligands have alternate binding site located between helices H2b, H3 and the β-sheets (Figure 1). PPARγ ligands that bind to this alternate site do not compete with endogenous ligands for their established orthosteric binding pocket, so this binding site can be identifies as an allosteric site. Allosteric binding residues include Glu259, Lys265, His266, Arg288, Ser289, Glu295, Ser342, Glu343, and Lys367. Several molecules have been reported to bind to the allosteric site [32,33]. It is found that the presence of an orthosteric ligand promotes the binding of the allosteric ligand [33].

Since PPARγ antagonists have potentials as anticancer agents, it is important to identify which binding site, orthosteric or allosteric, is preferred for PPARγ antagonists. The answer to this question and the structural basis for PPARγ antagonist binding will help future PPARγ antagonist drug discovery. To achieve this goal, we ran a series of docking studies of verified PPARγ antagonists with the crystal structures of PPARγ and analyzed the protein-ligand interactions to identity which residues are responsible for antagonist binding.

## 2. Computational Methods

### 2.1. Preparation of Protein Structures

We downloaded the wild type X-ray crystal structures of PPARγ complexed with *N*-sulfonyl-2-indole carboxamides (NSI) ligand (PDB ID: 2HFP) (Figure 1) from the RCSB Protein Data Bank (https://www.rcsb.org/structure/2hfp, accessed on 1 July 2021 [31]. There are no missing residues in the crystal structure of 2HFP. Structure of 2HFP was imported to Maestro program in the Schrödinger software suite. The Protein Preparation module was used for protein structure preparation. The pKas of protein side chains were calculated using PROPKA under pH 7.0. The side-chain structures of Gln, Asn and His were permitted to flip to maximize H-bond interaction. All lysines and arginines were protonated to +1 charge and all glutamic acid and aspartic acid were deprotonated to −1 charge. Histidine 425 were assigned as protonated HIP and histidines at 217, 266 and 466 were assigned as HIE while His323, Gln294 and Gln430 were flipped to maximized H-bond interactions. In addition, water molecules in the crystal structure were deleted before energy minimization. During the minimization, the backbone atoms were first restrained and then allowed to move freely using the OPLS force field in the MacroModel module in the Schrödinger software suite [34].

### 2.2. Preparation of Ligands

We collected forty-seven PPARγ antagonists (Appendix A) from different sources [27,30,31]. These antagonists were built and minimized using MOE program [35] based on the bound ligand, *N*-sulfonyl-2-indole carboxamides (NSI), in 2HFP as a template. In addition, we also built nine agonists (Appendix A) for analyzing the protein-ligand interactions for agonists. All PPARγ ligands were imported to the Maestro, subjected to the EPik calculation and the subsequent optimization by the MacroModel program using the OPLS2005 force field. In addition, we downloaded a database of 3D molecules from the National Cancer Institute (NCI) [36], and then four hundred and twenty-three drug-like molecules (obtained after filtration based on the Lipinski’s rule of five, calculated in the MOE program) were randomly selected from this database to assess the enrichment factor and to validate our docking method. The molecules were then subjected to EPik calculation and energy minimization using the MacroModel program. The EPik program predicted the pKa for all aromatic amines at 1.27–2.59, the pKa for oxazoline ring nitrogen at 1.22, and the pKa for imidazoline ring nitrogen at 2.29, suggesting neutral, non-protonated nitrogen atoms for these moieties. The pKa for pyridine was predicted to be 6.01–7.56, suggesting a neutral aromatic amine.

### 2.3. Molecular Dockin

Docking of the antagonists into the orthosteric and allosteric binding sites of PPARγ was completed using the Schrödinger software suite. We generated two grid files for the crystal structure of 2HFP using the Glide Grid Generation protocol with the two bound ligands as respective centroids with box size of 25 Å from the centroid, one for orthosteric binding site and the other for allosteric binding site of PPARγ. All 47 PPARγ antagonists were docked to each of the two grid files, and we also ran docking for the 423 National Cancer Institute (NCI) drug-like molecules to these two grid files to calculate the enrichment factor. During the docking process, the scaling factor for receptor van der Waals for the nonpolar atoms was set to 0.8 to allow for some flexibility of the receptor, and the precision was set as extra precision. All other parameters were used as defaults. The output docking scores were given as extra precision glide scores (GScore). The GScore is considered as a predicated binding affinity as well as a predicated free energy of binding (∆G_PRED_). The output ΔG_PRED_ was then related to the experimental ΔG_EXP_, calculated from the experimental IC_50_ (nM) using the following equation [37].
$${\Delta G}_{\mathrm{EXP}}\left(\mathrm{kcal}/\mathrm{mol}\right)=\mathrm{RT}\ln\;\left({\mathrm{IC}}_{50\;}\left(\mathrm{nM}\right)\times 10^{-9}\right)/1000$$

The Pearson’s R (Pearson’s correlation coefficient) was estimated using Microsoft Excel and the root-mean-square error and mean-absolute-error were measured from MatLab. 

To evaluate the effect of flexibility of binding pockets on ligand binding, we ran induced fit docking of nine agonists to the orthosteric site and the allosteric site using the Induced Fit Docking (IFD) program in the Schrödinger software suite. During the IFD, the bound ligands at the orthosteric and allosteric sites were set as the centroid for respective docking. Maximum number of poses was set to 20 for the first round of Glide Docking and to 5 for the Glide Redocking. The best docking score and the average score of the top five reported poses were tabulated and were compared to the results from the regular Glide Dock protocol. 

### 2.4. Protein-Ligand Interactions

From the docking output, each docked pose was analyzed with its bound protein and the interacting residues were tabulated to show which residues have high frequency in ligand binding, suggesting their essential roles for ligand binding.

## 3. Results and Discussion

### 3.1. Glide Docking

We ran docking experiments of 47 PPARγ antagonists against the two PPARγ binding sites to evaluate which is more predictable for the PPARγ antagonists, the orthosteric or the allosteric binding site. PPARγ antagonist docking scores are listed in Table 1. The glide docking shows that the predicted glide dock scores (GScore) at the allosteric binding site were in good agreement with the experimentally obtained values (ΔGexp), which was calculated from the IC_50_s using the equation as described in the Section 2. Table 2 shows that PPARγ antagonists bind preferably to the allosteric site rather than the orthosteric site in that the binding to the allosteric site produced binding affinity that was closer to the experimental values.

The Pearson’s correlation coefficient at allosteric binding site was 0.80 and the correlation R^2^ was 0.64 (Figure 2), better than the respective numbers at the orthosteric binding site (0.62, and 0.39, respectively). In addition, the average of the difference (or error) between ΔG_EXP_ and ΔG_PRED_ (ΔΔG), the mean of absolute error (MAE), and the root-mean-square (RMS) error for PPARγ antagonists docked at allosteric binding site were 1.08, 1.10, and 1.29, respectively whereas the respective numbers at the orthosteric binding site were ΔΔG (1.50), MAE (1.63), and RMSE (1.83) (Table 2). Taken together, our docking results indicate that glide docking is capable of predicting ligands binding affinity in different pocket sites of PPARγ. The glide docking reveals that the ΔG_PRED_ of PPARγ antagonists is more correlated with the ΔG_EXP_ of PPARγ antagonists at allosteric binding site, and the statistical results such as Pearson’s correlation coefficient, correlation R^2^_,_ ΔΔG, MAE, and RMSE further confirm the preference of the allosteric binding sites for PPARγ antagonists. Therefore, for future structure-based PPARγ antagonist design, it is more reasonable to use the allosteric site, rather than the orthosteric site.

Other than the correlation between the predicted binding affinity (GScore) and the experimentally obtained data (ΔGexp), we also used other methods to validate our docking programs and scoring functions. One commonly used method is whether the docking program can reproduce the native binding conformation. Docking methods are considered trustworthy when they can generate poses very close to the native conformation, i.e., with low Root Mean Square Deviation (RMSD) value from the known conformation (usually 1.5 or 2 Å depending on ligand size) [38]. The superposition of the Glide-generated docked pose for NSI and the native conformation in 2HFP revealed that the RMSD between these two poses is 0.22 Å, so the glide docking can successfully generate the native conformation. 

In addition, the enrichment factor (EF) is often used to evaluate the validity of a docking program. The EF is a general measurement of the efficiency of a docking program: the higher the EF, the more accurate the docking program [39,40]. The EF and the true hit% measure the concentration of active inhibitors in a specific subset divided (*x*%) by the concentration of active inhibitors in the database [41]. The EF can validate if a docking method can identity active compounds from the drug-like molecules. To evaluate the EF, in addition to the docking of 47 PPARγ antagonists to the binding sites, we also docked 423 drug-like compounds to the same binding sites. The Glide docking scores for the drug-like molecules at the allosteric site were listed in Appendix A. Among a total of 470 compounds in the database, only 47 were verified active compounds. Among the top 10% compounds in the total database (47), 33 were reported PPARγ antagonists, resulting in the EF score of 7.0 and hit rate (or true hit%) of 70% (Table 3), which suggested that the docking program was able to identify the active compounds. The docking scores of the 423 drug-like compounds against the 2HFP crystal structure are listed (Appendix A). 

### 3.2. Binding Mode of PPARγ Ligands at the Allosteric Binding Site

Forty-seven PPARγ antagonists and nine agonists were docked to the allosteric binding site and the orthosteric site to identity the binding difference between antagonists and agonists. After docking was completed, we first analyzed the protein-ligand interactions of antagonists and agonists at these two sites. Residues interacting with the antagonists at the allosteric site are listed in Table 4 and residues for the orthosteric antagonist binding are provided in Appendix A. We also tabulated the number of residues interacting with antagonists at the two sites and the frequency of interacting residues was calculated (Figure 3).

Both Table 4 and Figure 3 show that residues Lys265, Arg288, and Ser342 are the preferred residues for antagonist binding at the allosteric site. The crystal structure of the PPARγ /NSI complex at allosteric binding site reveals that residues Lys265 and Ser342 of PPARγ form H-bonds to the NSI ligand. In addition, Lys367 and Ser289 form H-bonds with SR1664, an PPARγ antagonist (Figure 4). In our docking study, the NSI ligand was docked to PPARγ and formed two H-bonds with Lys265 and Ser342, which agreed with what was observed in the native ligand in the X-ray crystal structure (Figure 4A). Our previous work identified seventeen PPARγ antagonists that showed single digit micromolar concentration in growth inhibition of the LNCaP cells and the residues responsible for PPARγ binding were Arg288 and Ser342 [22].

Other than the non-covalent PPARγ antagonists, some covalent PPARγ antagonists have been discovered. MMT-160 is a PPARγ antagonist with an alkyne function group that binds covalently to Cys285, a residue on helix 3 of the allosteric site [42].

Among all residues within the allosteric binding pocket, Lys265 and Ser342 appear to be the most important. In terms of agonist binding to the allosteric site, Ser342 interacted with 33% of agonists. Comparing to the 80% when it binds with antagonists, it is still reasonable to conclude that residues Lys265 and Ser342 are not only important for antagonist binding, but also can be utilized to design antagonists with selectivity over agonists. Residue Glu343 at the allosteric site appears to selectively bind to agonist (Figure 3). 

In addition to Lys265, Ser342, and Arg288, residues Lys367, Phe282 and Ser289 have been observed as binding residues to some PPARγ antagonists. However, they were not selective toward PPARγ antagonist binding. On the contrary, Lys367, Phe282 and Ser289 were more selective toward PPARγ agonist binding (Figure 3).

### 3.3. Binding Mode of PPARγ Ligands at the Orthosteric Binding Site

To identify which residues are important for agonist binding at the orthosteric binding site, we docked nine reported agonists to both the orthosteric and allosteric binding pockets. The structures of nine PPARγ agonists were given in Appendix A and were taken from published papers [43,44,45,46,47,48,49]. 

Table 5 shows that the docking scores for agonists at the orthosteric binding site in general were more negative (i.e., a stronger binding) than the same agonist at the allosteric site, which agreed with the fact that an agonist would prefer to bind to its target protein at the orthosteric site rather than the allosteric site. Residues responsible for orthosteric binding for an agonist are Leu228, Phe282, Lys367, His449 and Ser289 (Table 5 and Figure 3). It is interesting to observe that some of the agonists were able to bind to Ser342 and Glu343 of the allosteric site (Figure 3).

Figure 5 showed that Leu228, a PPARγ selective residue, as suggested in Figure 3, can provide main chain H-bond interactions with agonists and His449 provided an π-π interactions with aromatic rings on the ligand. 

As we discussed in the Section 1, the large PPARγ binding pocket allows the binding of two ligands, one at the orthosteric and the other at the allosteric binding site. Thus, it is not uncommon to find out that there are two molecules that occupy these two sites. The two molecules can be the same or different. When two of the same molecules bind to PPARγ, the ratio between ligand and PPARγ becomes 2:1, as what was observed in the crystal structure of 2HFP and 5UGM [31,49]. Brust et al. observed that an antagonist can bind to the orthosteric binding site and was still able to significantly reduce the cellular activation. However, dual-site binding can make an antagonist almost shut down the cellular activation [33]. Thus, it is not uncommon to observe that an antagonist can bind to the orthosteric site, which is what we observed in Table 1 and Appendix A. By inspecting at Table 1, one may find that some antagonists show stronger binding at the orthosteric site than at the allosteric site. However, as what we discussed in Section 3.1, antagonists should bind to the allosteric site as the binding at the allosteric site is more agreeable to what was observed from experiments. 

To design antagonists that can effectively compete with endogenous agonists, it is often true to introduce extra functional groups to the antagonists. Thus, antagonists normally have larger molecule weights, and more interacting surface. The average molecular weight (MW) of the nine agonists was 430.16 Dalton, whereas the mean MW of the 47 antagonists was 531.00 Dalton. To assess the effect of flexibility of binding pockets on ligand binding, we ran induced fit docking of nine agonists to the orthosteric site and the allosteric site using the Induced Fit Docking program in the Schrödinger software suite. 

By comparing the IFD-based docking scores to those from the Glide Dock, one can notice that these two sets of scores were overall very comparable with mean absolute error of 1.12 kcal/mol for the orthosteric site and 1.47 kcal/mol for the allosteric site (Table 6), although some large differences (greater than 2.0 kcal/mol) can be found in some ligands. This suggested that the way the Glide docking to implicitly incorporate the induced-fit effect by using the scaling factor for receptor van der Waals for the nonpolar atoms of 0.8 to allow for some flexibility of the receptor was sufficient in most cases. 

Table 6 also showed that the average docking scores of the top five poses between the agonists at the orthosteric site and the allosteric site were very close, with mean of the absolute errors of 0.61 kcal/mol. To find out what caused such a small difference for the same agonist that was docked from different allosteric and orthosteric site. We found out that the Induced-Fit Dock program was able to generate similar poses for the agonists, no matter they were docked from the orthosteric site or the allosteric site. The reason might be stemmed from the following two reasons, first, the large box size for docking (set at 25 Å from the centroid) covered both the orthosteric site and the allosteric site, which allowed the program to find their best binding conformation, though starting from a different centroid; second, all the agonists under study were able to adopt an extended conformation which allowed it to occupy both the orthosteric and allosteric sites. Figure 6 showed that the docking poses of the rosiglitazone and lobeglitazone, generated from two different centroids, occupied very similar conformational space, thus having very similar interactions with binding residues Ser289, Tyr327, His323, Lys367 and Tyr473 of the orthosteric site and residues Cys285, Lys265, Glu343, and Ser342 of the allosteric site. This is also in accordance with our Glide Dock observation that agonists were able to bind to Ser342 and Glu343 (Figure 3). The interacting residues of PPARγ agonists at the orthosteric and allosteric site based on the Induced-Fit Docking were listed in Appendix A. The crystal structure of the PPARγ/ lobeglitazone (PDB id: 5Y2T) [50] also showed that lobeglitazone can occupy both sites (Figure 1). Our docking pose of lobeglitazone agreed well with what was observed from the experiments. The antagonists from Hopkins’ laboratory [31], on the other hand, were able to adopt a curled conformation and thus preferentially bind to the allosteric site. 

## 4. Conclusions

PPARγ is an attractive target for drug discovery and development. PPARγ antagonists have showed antitumor activity against different tumors. The crystal structures of PPARγ complexed with antagonists revealed that antagonists can occupy two binding sites, orthosteric pocket, and allosteric pocket. Docking of PPARγ antagonists resulted in predicted binding affinity that was in good agreement with the experimentally observed binding affinity. Docking studies show that the experimental binding affinity of PPARγ antagonists is more correlated to the allosteric binding site than the orthosteric binding site. In addition, the statistics numbers of docking scores further confirmed the validity of the docking approach and favored the allosteric site. Therefore, the allosteric site is the most favorable binding site for PPARγ antagonists. Residues important for selective allosteric antagonism appeared to be Lys265, Ser342 and Arg288. Our data also suggested that the results from the Glide Dock in most cases were in good agreement with those of the Induced-Fit Docking.

## Figures and Tables

**Figure 1 biomolecules-12-01614-f001:**
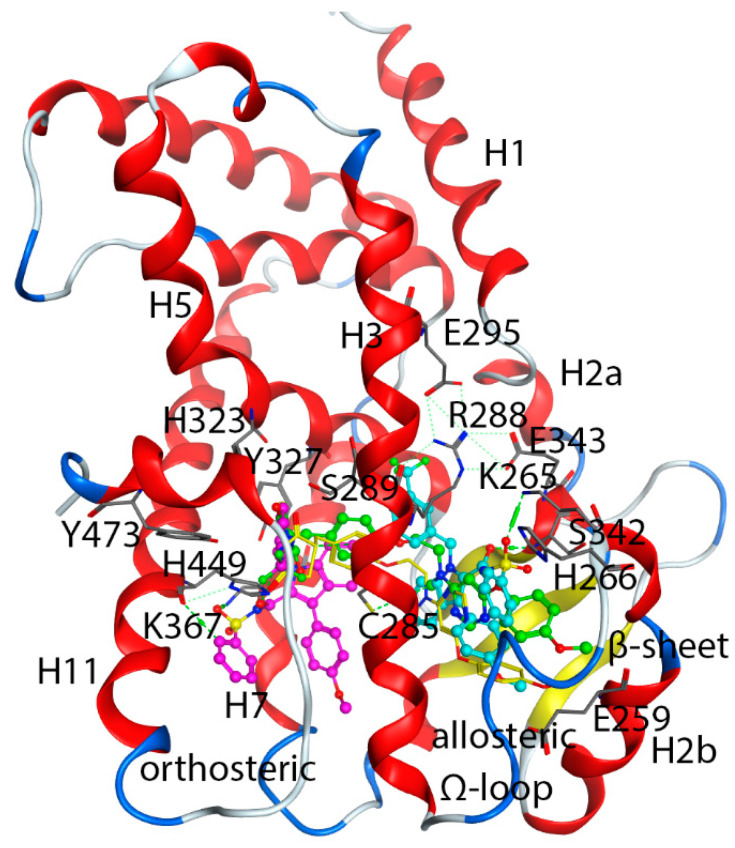
Co-crystal structure of NSI-bound PPARγ (PDB id: 2HFP). Among two bound ligands (PDB ligand ID: NSI), one binds to the orthosteric pocket (magenta NSI) and the other to the allosteric site (cyan NSI). Lobeglitazone of native bound structure (PDB id: 5Y2T), yellow; Lobeglitazone from the docked pose of this study, green. Some helices and interacting residues for the orthosteric and allosteric bindings were labeled. Residues were presented as one letter code for clarity.

**Figure 2 biomolecules-12-01614-f002:**
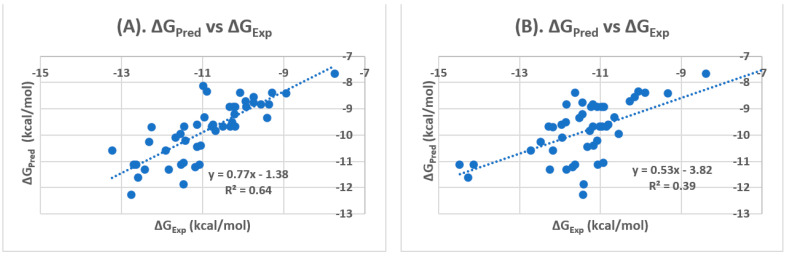
Correlations of glide docking scores for PPARγ antagonists: (**A**) Allosteric Binding site. (**B**) Orthosteric Binding Site.

**Figure 3 biomolecules-12-01614-f003:**
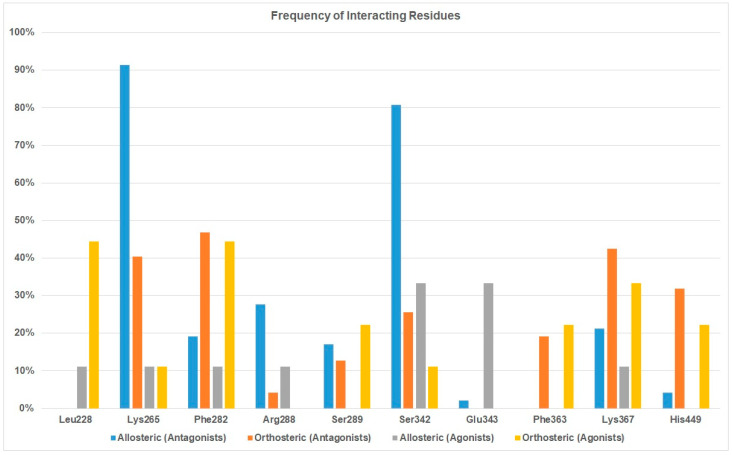
Frequency of interacting residues of PPARγ antagonists and agonists at the orthosteric and allosteric binding sites.

**Figure 4 biomolecules-12-01614-f004:**
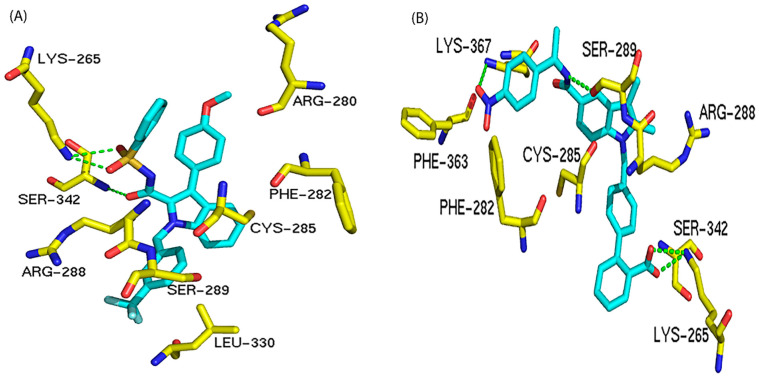
Interactions between PPARγ and different antagonists at the allosteric binding site. The H-bond interactions are depicted as green dotted lines. (**A**) Interactions between PPARγ and NSI (an antagonist). (**B**) Interactions between PPARγ and compound SR1664 (an antagonist).

**Figure 5 biomolecules-12-01614-f005:**
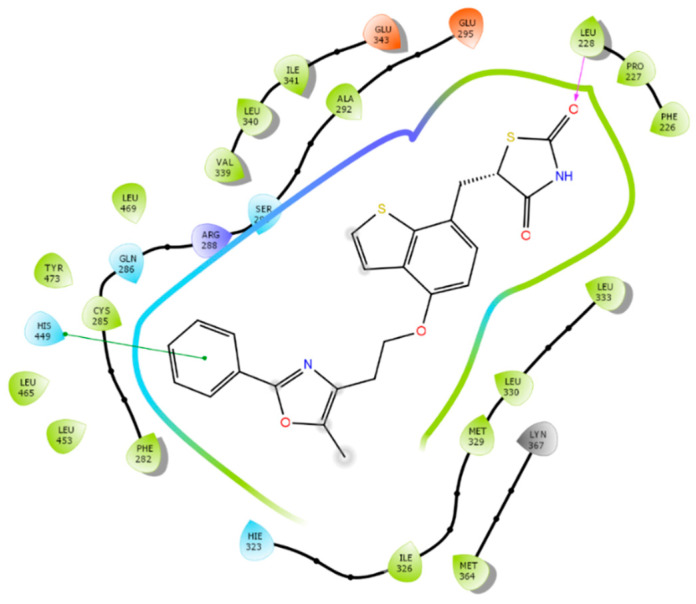
Interactions between PPARγ and edaglitazone (an agonist) at the orthosteric site. Pink arrow: H-bond interactions with Leu228 main chain; green line: π-π interactions between phenyl group of edaglitazone and the histidine aromatic ring.

**Figure 6 biomolecules-12-01614-f006:**
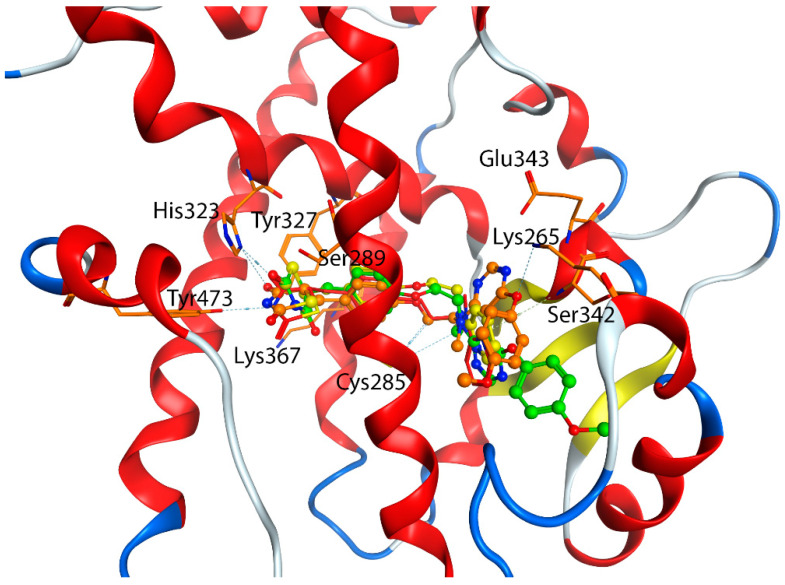
Interactions between PPARγ agonists at the orthosteric site and allosteric site based on the Induced-Fit Docking. Yellow: Rosiglitazone from the orthosteric site; Red: Rosiglitazone from the allosteric site; Green: Lobeglitazone from the orthosteric site; Orange: Lobeglitazone from the allosteric site.

**Table 1 biomolecules-12-01614-t001:** The glide score (kcal/mol) for the 47 PPARγ antagonists against the allosteric and orthosteric binding site.

Compound	IC_50_ (nM)	∆G_EXP_ (kcal/mol)	Allosteric GScore	Orthosteric GScore	Compound	IC_50_ (nM)	∆G_EXP_ (kcal/mol)	Allosteric GScore	Orthosteric GScore
**NSI**	3	−11.63	−12.59	−14.28	**22**	380	−8.76	−9.74	−11.45
**SR1664**	80	−9.68	−10.31	−10.96	**23**	330	−8.84	−9.37	−11.19
**SR11023**	108	−9.50	−10.27	−11.85	**24**	330	−8.84	−9.57	−11.84
**1**	7	−11.12	−12.64	−14.48	**25**	770	−8.34	−10.89	−10.06
**2**	1	−12.28	−12.75	−11.42	**26**	540	−8.55	−9.74	−10.15
**3**	7	−11.12	−12.69	−14.14	**27**	6	−11.22	−11.19	−11.68
**4**	2	−11.87	−11.46	−11.42	**28**	32	−10.22	−11.42	−11.08
**5**	290	−8.92	−10.33	−10.97	**29**	24	−10.39	−11.04	−11.17
**6**	720	−8.38	−10.07	−11.63	**30**	30	−10.26	−12.32	−12.48
**7**	280	−8.94	−10.25	−10.92	**31**	7	−11.12	−11.54	−11.63
**8**	80	−9.68	−10.78	−10.85	**32**	5	−11.32	−12.43	−12.25
**9**	290	−8.92	−10.19	−11.07	**33**	8	−11.05	−11.47	−10.93
**10**	180	−9.20	−10.21	−11.45	**34**	5	−11.32	−11.84	−11.83
**11**	90	−9.61	−11.14	−11.96	**35**	17	−10.60	−11.91	−12.73
**12**	80	−9.68	−10.50	−11.02	**36**	40	−10.09	−11.66	−11.95
**13**	80	−9.68	−10.20	−11.19	**37**	7	−11.12	−11.07	−11.06
**14**	680	−8.41	−8.93	−9.34	**38**	22	−10.45	−11.13	−11.32
**15**	700	−8.40	−9.28	−9.90	**39**	77	−9.70	−12.26	−12.17
**16**	140	−9.35	−9.40	−11.51	**40**	62	−9.83	−10.68	−11.27
**17**	90	−9.61	−10.74	−10.80	**41**	148	−9.32	−10.96	−10.66
**18**	2440	−7.66	−7.74	−8.39	**42**	17	−10.60	−13.23	−12.17
**19**	400	−8.73	−9.94	−10.27	**43**	1100	−8.13	−10.98	−6.61
**20**	50	−9.96	−11.55	−10.55	**44**	80	−9.68	−11.45	−12.28
**21**	280	−8.94	−9.93	−11.24					

**Table 2 biomolecules-12-01614-t002:** Statistical results, the correlation and residual errors between the experimentally obtained ∆G_EXP_ and the glide scores for the 47 PPARγ antagonists against the allosteric and orthosteric binding site.

	Allosteric Binding Site	Orthosteric Binding Site
**Pearson’s *R***	0.80	0.62
**Correlation *R*^2^**	0.64	0.39
**ΔΔG**	1.08	1.50
**MAE**	1.10	1.63
**RMSE**	1.29	1.83

**Table 3 biomolecules-12-01614-t003:** Enrichment factor of the glide docking against PPARγ at allosteric site.

Number of active PPARγ antagonists	47
Number of total compounds in the database	470
Number of active PPARγ antagonists in the top 10% subset	33
Enrichment factor (EF)	7.0
Hit rate (HR)	70%

**Table 4 biomolecules-12-01614-t004:** Interacting residues of PPARγ antagonists at the allosteric site.

Compound	Interacting Residues	Compound	Interacting Residues
**NSI**	Lys265, Ser342	**22**	Lys265, Ser342
**SR1664**	Lys265, Ser289, Ser342, Lys367	**23**	Lys265, Arg288, Ser342, Lys367
**SR11023**	Lys265, Ser289	**24**	Lys265, Arg288, Ser342
**1**	Lys265, Ser342	**25**	Lys265, Arg288, Ser342
**2**	Lys265, Ser342	**26**	Lys265, Arg288, Ser342
**3**	Lys265, Arg288, Ser342	**27**	Lys265, Ser289, Ser342
**4**	Lys265, Ser342	**28**	Lys265, Ser289, Ser342
**5**	Lys265, Ser342	**29**	Lys265, Ser289
**6**	Lys265, Ser342	**30**	Lys265, Phe282, Ser342, Lys367
**7**	Lys265, Ser342	**31**	Lys265, Ser289
**8**	Lys265, Arg288, Ser342	**32**	Lys265, Phe282, Ser342, Lys367
**9**	Lys265, Ser342	**33**	Lys265, Ser289
**10**	Lys265, Arg288, Ser342	**34**	Lys265, Ser289, Ser342, His449
**11**	Lys265, Arg288, Ser342	**35**	Lys265, Phe282, Ser342, Lys367, His449
**12**	Lys265, Ser342	**36**	Lys265, Phe282, Lys367
**13**	Lys265, Ser342	**37**	Lys265, Ser342
**14**	Lys265, Arg288, Ser342	**38**	Phe282, Lys367
**15**	Lys265, Ser342	**39**	Lys265, Phe282, Lys367
**16**	Lys265, Arg288, Ser342	**40**	Lys265, Gly284
**17**	Lys265, Ser342	**41**	Phe282
**18**	Ser342	**42**	Lys265, Phe282, Ser342, Lys367
**19**	Lys265, Arg288, Ser342	**43**	Arg288
**20**	Lys265, Ser342	**44**	Lys265, Phe282, Ser342, Lys367
**21**	Arg288, Ser342, Glu343		

**Table 5 biomolecules-12-01614-t005:** The glide scores (kcal/mol) and interacting residues for the nine PPARγ agonists against the allosteric and orthosteric binding site.

Names	Orthosteric GScore	Allosteric GScore	Orthosteric	Allosteric
**Rosiglitazone**	−8.45	−6.82	Leu228	NA
**Lobeglitazone**	−9.70	−8.56	Leu228, Phe282	Leu228
**Pioglitazone**	−8.96	−2.72	Ser289, Tyr327, Leu228	Ser342, Leu340
**GW1929**	−12.00	−9.93	Phe282, Phe363, Lys367	Lys265, Arg288
**Farglitazar**	−11.85	−12.58	Phe282, Lys367, Phe363	Lys367, Phe282
**Edaglitazone**	−9.98	−9.68	Leu228, His449	Glu343
**Amorfrutin 1**	−8.74	−8.42	Ser289 (phenol)	Glu343, Leu340
**Amorfrutin 2**	−7.89	−7.91	Phe282 (pp), Lys367	Glu343, Ser342
**MRL20**	−7.90	−9.72	Ser342, His449	Lys265, Ser342

**Table 6 biomolecules-12-01614-t006:** The glide scores (kcal/mol) (the best docked pose and the average of the top five poses) and the absolute errors (AE) between the average top 5 IFD-glide scores and the respective Glide Dock glide scores for the nine PPARγ agonists against the allosteric and orthosteric binding site.

Names	Ortho. Best	Ortho. Mean	Ortho. AE	Allo. Best	Allo. Mean	Allo. AE
**Rosiglitazone**	−11.74	−8.71	0.26	−9.80	−9.05	2.22
**Lobeglitazone**	−12.14	−10.39	0.69	−11.65	−10.95	2.39
**Pioglitazone**	−8.62	−6.34	2.61	−9.00	−6.83	4.11
**GW1929**	−11.01	−9.65	2.35	−11.73	−8.88	1.05
**Farglitazar**	−11.61	−10.79	1.05	−13.48	−12.54	0.04
**Edaglitazone**	−9.30	−8.59	1.39	−11.06	−9.86	0.18
**Amorfrutin 1**	−8.97	−8.44	0.30	−8.94	−8.60	0.18
**Amorfrutin 2**	−7.96	−7.66	0.23	−8.14	−7.67	0.24
**MRL20**	−9.57	−6.72	1.18	−9.91	−6.89	2.82
**MAE**			1.12			1.47

Note: Abbreviation: Ortho. orthosteric; Allo. allosteric.

## Data Availability

Not applicable.

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
