# Peer review of "Molecular Modeling of Allosteric Site of Isoform-Specific Inhibition of the Peroxisome Proliferator-Activated Receptor PPARγ"

_biomolecules, 2022, doi:10.3390/biom12111614_

Round 1
Reviewer 1 Report
1. In "The effects of rosiglitazone on incident ma-lignancies were 0.88 in gastro-intestinal..." insert the acronym of gastro-intestinal (GI) used in the abridged version in the next sentence.
2. "The PPARγ ligand binding domain has a large, hydrophobic, and T-shaped cavity which allows it to accommodate two ligands. expands from β-sheet strands to car-boxy-terminal domain". The sentence is not clear: rephrase it.
3. thiazolidinedione: insert the acronym TZD
4. "All 47 PPARγ antagonists were docked to each of the two grid files, and we later run docking for the 423 NCI drug-like molecules were docked to the allosteric binding site of PPARγ grid file". The sentence is not clear: rephrase it.
5. In the paragraph 3.1 eliminate "In other words, it is more correlated" before The Pearson's...
6. In the paragraph 3.2 "Forty-seven PPARγ antagonists and nine agonists was docked". Was to replace with were.
7. It is not clear why anticancer drugs are argued in the introduction and the binding mode of PPARy receptor was performed using SR1664, an anti-diabetic drug.
Author Response
The authors would like to thank the reviewers for their constructive and helpful feedbacks. We have addressed all the requests and revise our manuscript accordingly. The revisions are highlighted in cyan.
- In "The effects of rosiglitazone on incident ma-lignancies were 0.88 in gastro-intestinal..." insert the acronym of gastro-intestinal (GI) used in the abridged version in the next sentence.
Fixed.
- "The PPARγ ligand binding domain has a large, hydrophobic, and T-shaped cavity which allows it to accommodate two ligands. expandsfrom β-sheet strands to car-boxy-terminal domain". The sentence is not clear: rephrase it.
Fixed, by deleting “expands from β-sheet strands to car-boxy-terminal domain”
- thiazolidinedione: insert the acronym TZD
Fixed
- "All 47 PPARγ antagonists were docked to each of the two grid files, and we later run docking for the 423 NCI drug-like molecules were docked to the allosteric binding site of PPARγ grid file". The sentence is not clear: rephrase it.
Fixed, see section 2.3 highlighted changes.
- In the paragraph 3.1 eliminate "In other words, it is more correlated" before The Pearson's...
We deleted this sentence.
- In the paragraph 3.2 "Forty-seven PPARγ antagonists and nine agonists wasdocked". Was to replace with were.
Fixed.
- It is not clear why anticancer drugs are argued in the introduction and the binding mode of PPARy receptor was performed using SR1664, an anti-diabetic drug.
Fixed. SR1664 is an PPARγ antagonist, since this paper is to elucidate the binding mode of PPARγ antagonists, thus it is reasonable to include SR1664 in this study. We added “(such as SR1664)” after “Several PPARγ antagonists” (page 2, highlighted sentences) and add a sentence, “They have potentials to be anticancer or antidiabetic drugs”
Reviewer 2 Report
*The reviewer thinks modeling is only a speculation. The authors should disclose the detailed experimental evidence to bind directly one antagonist to the allosteric binding site of PPARgamma antagonist.
*The authors should include the results of MMT-160 ( Bioorg Med Chem Lett. 2022 May 15;64:128676).
Author Response
The authors would like to thank the reviewers for their constructive and helpful feedbacks. We have addressed all the requests and revise our manuscript accordingly. The revisions are highlighted in cyan.
*The reviewer thinks modeling is only a speculation. The authors should disclose the detailed experimental evidence to bind directly one antagonist to the allosteric binding site of PPARgamma antagonist.
The authors agreed with the reviewer’s point that “modeling is only” theoretical. But it is not mere speculation. It is always a good practice to validate the modeling method by comparing the modeling results to the experimental results. We believe we have convinced the readers in section 3.1, both the discussion and the data (Fig. 2 and Tables 1, 2 and Table 3), that our data supported the validity of the modeling method. To address the experimental evidence, section 3.2 highlighted the experimentally-obtained PPARγ/antagonist interactions at the allosteric site (also Fig. 4). We also added a sentence of summary regarding our previous experimental work on identifying PPARγ antagonists (highlighted in cyan).
*The authors should include the results of MMT-160 ( Bioorg Med Chem Lett. 2022 May 15;64:128676).
This paper has been cited as Ref. #42, and related discussion was added in Section 3.1, just before Table 4.
Reviewer 3 Report
The manuscript by Almahmoud et.al., models the allosteric Site of Isoform-Specific Inhibition of the Peroxisome Proliferator-activated Receptor PPARγ. This publication highlights that PPARγ antagonists such as antidiabetic drugs interact more efficiently at the allosteric site than the orthosteric site. The work may be publishable after the below concerns are addressed.
Major comments
1) The molecular sizes of agonists and antagonists considered in this study are not in the same range meaning that some are smaller and others are larger. How does the binding pocket adjust to various sizes of ligand? Why did the author not consider the induced fit docking to accommodate the flexibility of binding pockets with respect to various ligands?
2) It would be great if authors could include the results of the Epik calculations regarding pKa prediction and tautomerization of ligands in the 2.2 preparation of ligands of computational methods section.
3) The protein preparation section in 2.1 didn’t show any details of pka prediction of protein residues. I would request the authors to include the pka calculations for protein residues or if they didn’t perform please explain the reason for considering the pka calculations for protein residues given there are a couple of titratable residues in both the binding sites?
Minor comments
1) It would be easy to follow if authors define the residues of both allosteric and orthosteric binding sites in Figure 1.
2) The following sentence is repeated twice at the end of the abstract section and the conclusion section. I would suggest the authors modify the sentence in one of the section and reduce the repetitiveness.
“ The PPARγ antagonists seem to selectively bind to residues Lys265, Ser342 and Arg288 at the allosteric binding site, whereas PPARγ agonists would selectively bind to residues Leu228, Phe363, and His449, though Phe282 and Lys367 may also play a role for agonist binding at the orthosteric binding pocket. This finding will provide new perspectives in the design and optimization of selective and potent PPARγ antagonists or agonists”
Author Response
The authors would like to thank the reviewers for their constructive and helpful feedbacks. We have addressed all the requests and revise our manuscript accordingly. The revisions are highlighted in cyan.
The manuscript by Almahmoud et.al., models the allosteric Site of Isoform-Specific Inhibition of the Peroxisome Proliferator-activated Receptor PPARγ. This publication highlights that PPARγ antagonists such as antidiabetic drugs interact more efficiently at the allosteric site than the orthosteric site. The work may be publishable after the below concerns are addressed.
Major comments
1) The molecular sizes of agonists and antagonists considered in this study are not in the same range meaning that some are smaller and others are larger. How does the binding pocket adjust to various sizes of ligand? Why did the author not consider the induced fit docking to accommodate the flexibility of binding pockets with respect to various ligands?
It is true that antagonists normally have larger molecule weights than the agonists. Antagonists with larger molecular weight would have larger interacting surface and thus better binding to the protein. Proteins were able to accommodate the ligands by induced-fit binding. To assess the effect of flexibility of binding pockets on ligand binding, we ran induced fit docking of nine agonists to the orthosteric site and the allosteric site using the Induced Fit Docking program in the Schrödinger software suite. Please see section 2.3 (highlighted in cyan) in method and section 3.3 (highlighted in cyan, before conclusion) for discussion.
2) It would be great if authors could include the results of the Epik calculations regarding pKa prediction and tautomerization of ligands in the 2.2 preparation of ligands of computational methods section.
We thank the reviewer raised the important issue of ionization based on pKa prediction. We did run Epik program on ligands before docking study as specified in section 2.2. To make it clearer, we have added a few sentences in section 2.2 (highlighted in cyan)
3) The protein preparation section in 2.1 didn’t show any details of pka prediction of protein residues. I would request the authors to include the pka calculations for protein residues or if they didn’t perform please explain the reason for considering the pka calculations for protein residues given there are a couple of titratable residues in both the binding sites?
We thank the reviewer raised the important issue of ionization based on pKa prediction. Actually, the protein preparation module in the Schrödinger Maestro program has incorporated the pka prediction and thus adjusted the ionization state of residues according. We have added a few sentences in section 2.1 (highlighted in cyan) to make it clearer.
Minor comments
- It would be easy to follow if authors define the residues of both allosteric and orthosteric binding sites in Figure 1.
Fixed. Figure 1 were updated with labeled residues.
2) The following sentence is repeated twice at the end of the abstract section and the conclusion section. I would suggest the authors modify the sentence in one of the section and reduce the repetitiveness.
“ The PPARγ antagonists seem to selectively bind to residues Lys265, Ser342 and Arg288 at the allosteric binding site, whereas PPARγ agonists would selectively bind to residues Leu228, Phe363, and His449, though Phe282 and Lys367 may also play a role for agonist binding at the orthosteric binding pocket. This finding will provide new perspectives in the design and optimization of selective and potent PPARγ antagonists or agonists”
We have revised the conclusion (highlighted in cyan).
Round 2
Reviewer 2 Report
The reviewer thinks this revised manuscript satisfactorily revised.